# Kernel Partial Least Squares Feature Selection Based on Maximum Weight Minimum Redundancy

**DOI:** 10.3390/e25020325

**Published:** 2023-02-10

**Authors:** Xiling Liu, Shuisheng Zhou

**Affiliations:** 1School of Mathematics and Statistics, Xidian University, Xi’an 710071, China; 2Public Education Department, Zhengzhou University of Economics and Business, Zhengzhou 451191, China

**Keywords:** feature selection, Relief, maximum weight minimum redundancy, kernel partial least squares

## Abstract

Feature selection refers to a vital function in machine learning and data mining. The maximum weight minimum redundancy feature selection method not only considers the importance of features but also reduces the redundancy among features. However, the characteristics of various datasets are not identical, and thus the feature selection method should have different feature evaluation criteria for all datasets. Additionally, high-dimensional data analysis poses a challenge to enhancing the classification performance of the different feature selection methods. This study presents a kernel partial least squares feature selection method on the basis of the enhanced maximum weight minimum redundancy algorithm to simplify the calculation and improve the classification accuracy of high-dimensional datasets. By introducing a weight factor, the correlation between the maximum weight and the minimum redundancy in the evaluation criterion can be adjusted to develop an improved maximum weight minimum redundancy method. In this study, the proposed KPLS feature selection method considers the redundancy between the features and the feature weighting between any feature and a class label in different datasets. Moreover, the feature selection method proposed in this study has been tested regarding its classification accuracy on data containing noise and several datasets. The experimental findings achieved using different datasets explore the feasibility and effectiveness of the proposed method which can select an optimal feature subset and obtain great classification performance based on three different metrics when compared with other feature selection methods.

## 1. Introduction

Feature selection is a kind of critical issue in machine learning that aims to identify the optimal feature subset by removing irrelevant or redundant features and enhancing the accuracy of classification [1]. Feature selection has been extensively applied in image recognition [2], text classification [3], image retrieval [4], fault diagnosis [5], bioinformatics data analysis [6], and so on. Nowadays, feature selection methods are mainly categorized into the following three main types: filter, wrapper, and embedded [1,7,8]. Filter methods [8,9] are independent of the subsequent learning algorithms. In the above-mentioned methods, the statistical performance evaluation characteristics of all training data are generally directly employed, which is a fast process, but the evaluation results show a large deviation from the performance of the subsequent learning algorithms. Wrapper methods [9,10] adopt the training accuracy of the subsequent learning algorithm for evaluating the feature subset, which is inappropriate for large datasets due to its small deviation and the large amount of calculation involved [1]. Embedded methods [11] combine the feature selection training process with the feature learning algorithms, and the features are automatically selected while the model training is completed. However, embedded methods are insufficient for special algorithms [12]. Relative to the wrapper and embedded methods, the filter methods have been extensively applied in feature selection.

Feature weighting represents the degree of association between any feature and a class label [13]. The larger the feature weighting value, the stronger the classification ability of the feature. Thus, feature weighting can reflect the classification discrimination ability of any feature in a dataset. To date, numerous feature weighting methods have been developed for feature selection on the basis of various strategies. The Pearson correlation coefficient (PCC) method [1,14], based on features and classes, is a widely used statistical method for feature weighting. However, this method can only be employed for continuous variables, and these variables have line dependency. To counteract this drawback, Kendall’s rank correlation coefficient has been adopted for feature weighting [15]. Mutual information [16,17] and information gain [18] methods that are on the basis of information theory have also been developed. The Relief method [14,19] based on feature weighting has been proposed as a classification model. This method can analyze a sample and its nearest neighbors from different classifications and the same classification. Thus, the importance of each sample feature of the dataset can be demonstrated. The ReliefF method [20], as an extension of the Relief algorithm, can describe multilabel feature selections. Moreover, the Fisher score (FS) method [21] can be regarded as a supervised feature weighting method, representing the recognition capability of each sample feature depending on its calculated Fisher score. The Laplacian score [22], as an unsupervised feature weighting method, can analyze the variance and local retention of each feature, reflecting the capability of each feature to retain its original data manifold structure. Lastly, the Constraint score [23] can be applied in semi-supervised and supervised learning by calculating the pair-wise constraints between the data samples as the feature weighting. In addition to the above methods, there are many other feature weighting methods. The reader can refer to the literature to learn the details of these methods [1,2,3,8].

Redundancy represents the degree of correlation of any one feature with another in a dataset. Minimum redundancy refers to the description of the dependence association between features and requires minimum correlation between features. Thus, features are selected by optimizing a special cost function, which is nothing but a trade-off between the largest amount of information about the characteristics and the least redundancy between any two features. The correlation-based feature selection (CFS) method [24] exploits the heuristic method with the purpose of assessing the value of a feature subset, which is on the basis of the assumption that the features included in the optimal feature subset need to be highly related to the classification, whereas the selected features are irrelevant. However, the fast correlation-based feature selection (Fast CFS) method [25] can lower the computational complexity of the CFS algorithm by eliminating the redundant features in the selected feature subset. Moreover, the minimum redundancy maximum relevance (mRMR) method [26] adopts mutual information for measuring the redundancy and relevance between the features. Further, two cost functions on the basis of mutual information are constructed to acquire the optimal feature subset. However, the CFS, Fast CFS, and mRMR methods can only be used for supervised learning tasks, and the number of selected features cannot be provided in advance in the filter methods of space search. In addition, the maximum weight and minimum redundancy (MWMR) method [27] has been developed to search for the optimal feature subsets in accordance with maximum weight and minimum redundancy. Moreover, the weight of each feature represents its importance, and the redundancy reflects the relationship between the features. However, due to the differences in the redundancy and feature weighting of different datasets, the MWMR method cannot efficiently consider the redundancy among features and the feature weighting between the feature and class labels of different datasets.

In addition, the various feature selection methods described above can efficiently handle processing in linear systems, which makes them unable to take advantage of the nonlinear relationships between variables. The partial least squares (PLS) method [28], as a set of techniques, can map the input and output variables into a new space with the overall objective of maximizing the covariance. The PLS method has been successfully adopted in various applications of machine learning, however, this method is only suitable for feature selection in linear systems. As an extension of PLS, the kernel partial least squares (KPLS) method [29], can address the nonlinear systems better than the PLS method. KPLS can be adopted for selecting an optimal set of features. The kernel method employs a kernel function that conforms to the inner product of the feature space, which avoids nonlinear optimization [30,31]. Thus, the KPLS method is an effective and fast method to process nonlinear systems and can be adopted for small and large datasets.

In the current work, a KPLS feature selection method on the basis of the improved MWMR has been proposed. The method proposed in this study applies the advantages of the KPLS feature selection method in nonlinear systems by using the ReliefF algorithm to calculate the dependency between a feature and a class label. As an improvement to the correlation of the maximum weight and minimum redundancy in the MWMR [27] method, the method adjusts the ratio of the maximum weight and minimum redundancy in the evaluation criteria of different datasets by adding a weight factor. To learn the classification performance of this method on different datasets, the proposed feature selection method has been tested in line with the classification accuracy using a linear support vector machine (SVM) classifier [32]. In order to obtain classification accuracy, the performance of the proposed method has been compared with the FS [21], CFS [24], ReliefF [20], and mRMR [27] methods by using a 10-fold cross-validation 10 times.

The remainder of this study has been structured as follows: The concept of PLS, KPLS, and the related feature selection methods are shown in Section 2, the proposed KPLS feature selection method is depicted in Section 3, the experimental findings are presented in Section 4, and conclusions from this study and some future directions are illustrated in Section 5.

## 2. KPLS Method and Feature Selection

In the current section, we first introduce the PLS and KPLS methods, and then describe the improved MWMR algorithm and discuss its useful properties.

### 2.1. PLS Method

The PLS method has been proven to be a valuable and popular method for modeling the correlation between two datasets. This method can also be applied in dimension reduction techniques and modeling, as well as classification and regression.

Consider a pair of input data, $X\in R^{n\times m}$, with *n* samples and *m* features and output data, $Y\in R^{n\times q}$, with *n* samples and *q* features. The PLS method mainly aims to maximize the association between the input data, *X*, and the output data, *Y*, using an iterative method, and applies least squares regression on the principal components. Thus, the PLS model describes the maximum correlation between the input and output data, and can be adopted for finding the approach to the following optimization problem [28]: (1)$$\begin{array}{l} \underset{w\in R^{m},c\in R^{q}}{\max}<Xw,Yc>\\ s.t.w^{T}w=1,c^{T}c=1 \end{array}$$
where *w* and *c* represent the weight vectors of *X* and *Y*, respectively. This iterative process can continue until a stop condition is satisfied [28].

### 2.2. KPLS Method

The KPLS method [29] was proposed to measure the nonlinear correlation in the kernel space. In this method, a nonlinear multiple regression model is constructed between the input and output variables. In addition, the original input variables are first transformed into a high-dimensional or even infinite-dimensional feature space, and subsequently, the linear PLS model is built in this high-dimensional feature space [29,33]. The KPLS method avoids nonlinear optimization of the feature space by using kernel functions. Therefore, as an iterative algorithm, the KPLS model can be introduced to process the feature selection problem in low- and high-dimensional datasets. 

For a pair of input data, $X\in R^{n\times m}$, with *n* samples and *m* features, output data, $Y\in R^{n\times q}$, with *n* samples and *q* features, and a mapping function, $\varphi (x):R^{m}\to F$, the input dataset, *X,* is mapped into a reproducing kernel Hilbert space, in which the target space can be extremely large or even infinite-dimensional. Therefore, *X* and *Y* are transformed into the feature matrices $\Phi (X)=[\varphi (x_{1}),\varphi (x_{2}),\cdots ,\varphi (x_{n})]$ and $\Phi (Y)=[\varphi (y_{1}),\varphi (y_{2}),\cdots ,\varphi (y_{n})]$, respectively. The KPLS method aims to find projection matrices maximizing the association between the input and output datasets. Therefore, the target function of KPLS can be expressed as [33]:(2)$$\begin{array}{l} \underset{w\in R^{s},c\in R^{t}}{\max}<\Phi (X)w,\Phi (Y)c>\\ s.t.w^{T}w=1,c^{T}c=1 \end{array}$$
where $K_{x}=\Phi (x)\Phi^{T}(x)\in R^{n\times n}$ refers to a kernel Gram matrix. Based on the kernel kick, this study avoids explicitly mapping the dataset into a high-dimensional feature space. The detailed iteration process of the KPLS algorithm can be found in previously published articles [29,33].

### 2.3. Improved MWMR Method

Weight (feature-class) and redundancy (feature-feature) analysis are the foundation of the MWMR [27] method. Feature weighting is adopted for calculating the ranking of the sample features based on the class labels, and relevance is used to calculate the redundancy between the features of the sample dataset. To shorten the training time and clearly display the correlation between the features in high-dimensional datasets, it is essential to choose features with a high classification discrimination ability, which is the maximum weight criterion. Further, to remove the redundant features from the dataset, two highly correlated features should be avoided from appearing in the selected feature subsets, which is referred to as the minimum redundancy criterion. The detailed maximum weight and minimum redundancy criteria are explained below.

The maximum weight (*MW*) between a feature and a class label can be acquired according to the feature weighting score (*W*_*score*(*f_i_*|*c*)), which is a number based on feature weighting that describes the ranking of a feature, *f_i_*, with respect to its class label, *c*. The *MW* can be written as
(3)$$MW=\underset{i}{\max}\{W\_score(f_{i}\left|c\right.)\}$$

The feature with the highest *W*_*score*(*f_i_*|*c*) has the strongest classification ability, whereas that with the lowest *W*_*score*(*f_i_*|*c*) has the lowest classification ability. Therefore, the feature with the maximum weight should be selected and that with the minimum weight should be deleted. This feature selection criterion is the *MW* criterion.

The minimum redundancy (*MR*) [34] between the features can be obtained with the feature relevance score (*R*_*score*(*f_i_*|*f_j_*)), which is on the basis of the correlation between features that describes the relevance of a selected feature, *f_i_*, with respect to a non-selected feature, *f_j_*. Thus, let *F* be the set of all the features and *F** be the set of the selected features. Then, *MR* can be written as
(4)$$MR=\underset{i,j}{\min}\{R\_score(f_{i}\left|f_{j}\right.)\}$$
where $f_{i}\in F^{*};f_{j}\in S,S=F-F^{*}$. When two features are highly correlated, if one of them is discarded, the classification ability of the remaining features does not change much. Therefore, a non-selected feature and a selected feature with high redundancy can be deleted. Instead, the non-selected features and the selected feature with minimum redundancy should be selected. This feature selection criterion is the *MR* criterion [26].

The criterion used for determining the feature subset through the optimization of the conditions in Equations (3) and (4) simultaneously is called MWMR [27]. However, different datasets might lead to differences in the classification performance of the method used. Some datasets have more redundancy among their features, whereas others have more feature weighting between their features and their class labels. Therefore, if the redundancy and feature weighting in the evaluation criterion can be given different weight factors according to the internal attribute relations of the datasets, the features of the sample datasets can be learned accurately, and the effectiveness of feature selection can be improved. Thus, a weight factor, α, can be introduced such that the weight factor of the *MW* in the feature evaluation criterion is α and the weight factor of the MR is (1−α). During feature selection of different datasets, when the *MW* of a dataset plays a leading role, the value α corresponding to the *MW* is increased. On the contrary, the value (1−α), corresponding to the *MR*, is increased so that the *MR* plays a greater role in the selected feature subset. Thus, we can describe the improved MWMR model as:(5)$$R=\underset{}{\max}\{\alpha MW-(1-\alpha )\times MR)\}$$
where α is the weight factor that has been assigned to the *MW* in the optimizing criterion, R. In addition, the value of α has been tested to be between {0,1}. This range begins with α=0, which corresponds to Equation (4), and ends with α=1, corresponding to Equation (3). When α=0, the optimal feature selection criterion corresponds to choosing the optimal feature subset only in accordance with the minimum redundancy between the features. When α=1, the optimal criterion for feature selection corresponds to choosing the optimal feature subset only in accordance with the maximum weight between the feature and its class label. The optimal value of α has been determined by the 10-fold cross-validation classification accuracy [35].

## 3. Proposed KPLS Feature Selection on the Basis of the Improved MWMR Method

In the current work, our objective is to provide an efficient feature selection method for selecting the optimal feature subset, thus improving the speed and accuracy of the subsequent data classification process.

The MWMR algorithm [27] not only takes the association between feature weighting and class labels into account but also the redundancy between the features; additionally, it has high computational efficiency and is suitable for processing high-dimensional or large-scale datasets. However, in practice, some features in the datasets are strongly related to their class labels, whereas others are highly redundant. Therefore, the effectiveness of feature selection can be improved by assigning feature weighting and redundancy in the evaluation criterion depending on the internal attribute correlation of the datasets. On the other hand, the MWMR method [27] does not take into consideration the different correlations among feature weighting, redundancy, and classification in different datasets. To overcome these problems of this method, we have proposed an improved MWMR method, as described in Section 2. This has been achieved by introducing a weight factor, α, into the evaluation criterion such that the weight factor of the MW in the feature evaluation standard is α, and the weight factor of the *MR* is (1−α). Feature selection can then be performed for different datasets, α can be increased when the *MW* of the datasets exerts a dominant function, and the value of α corresponding to the *MR* can be increased conversely. In addition, the KPLS method can efficiently avoid nonlinear optimization problems with the use of kernel functions. This method can map the data to latent vectors and subsequently employ linear regression to those components, making it usable for both small and large-scale data.

Therefore, the present study shows a KPLS feature selection method in accordance with the improved MWMR (KPLS-MWMR) method. It is shown that the proposed feature selection method can calculate and enhance the classification accuracy of low-dimensional as well as high-dimensional datasets. When identifying a set to be the optimal feature subset, it not only considers the correlation between feature weighting and redundancy but also makes use of the advantages of the KPLS method to avoid nonlinear optimization. The proposed KPLS-MWMR method contains the following five basic steps: (1)Calculation of the latent matrix using the KPLS algorithm;(2)Calculation of the feature weighting score (*W*_*score*(*f_i_*|*c*)) based on the feature, *f_i_*, and the class label, *c*, of the dataset;(3)Calculation of the feature redundancy score (*R*_*score*(*f_i_*|*f_j_*)) based on the features *f_i_* and *f_j_* of the dataset;(4)Calculation of the objective function, R, according to the feature weighting score and feature redundancy score;(5)Selection of an optimal feature subset on the basis of the objective function, R. 

The flow chart of the proposed KPLS-MWMR feature selection method including the basic steps is shown in Figure 1 and the proposed KPLS-MWMR feature selection method is presented in Algorithm 1. First, the latent matrix is computed using the KPLS algorithm [29]. Second, the feature weighting scores for the sample dataset are generated using the ReliefF algorithm [20]. The characteristics of the sample dataset are arranged in descending order in accordance with the feature weighting scores, and the feature with the maximum feature weighting score in the dataset is selected. Third, the redundancy score between the selected feature and the non-selected features of the dataset is obtained using the PCC algorithm [14]. Finally, based on the features and the class labels, the objective function, R, can be calculated, and the optimal feature subset is chosen.
**Algorithm 1.** KPLS based on maximum weight minimum redundancy (KPLS-MWMR). Input: Feature dataset, X∈Rn×m, class label, Y∈Rn×1, feature number, *k*,weight factor, α.Output: A selected feature subset, *F**.(1) Initialize the feature dataset, *F*;(2) Let the feature set F*=∅;(3) Calculate the latent matrix: *F* = KPLS(*X*,*Y*) using the KPLS algorithm [29];(4) Calculate the feature weighting score: *W_score* (*F|Y*) using the Relief *F* algorithm [20];(5) Arrange the feature weighting score in descending order: [WS, rank] = descend(*W_score*(*F|Y*));(6) Form a feature subset *S = X*(:, rank);(7) Select the optimal feature subset *F** = S(:, 1);(8) For each *j* < *k;*(9) *f*_1_ = *S*(:, *j*);(10) *w* = WS(:, *j*);(11) Compute the feature redundancy score, *r = R*_*score*(*f*_1_|(S − *F**)) using the PCC algorithm [15];(12) Calculate the evaluation criteria, R: R=αw−(1−α)r according to Equation (5);(13) Arrange the values of *R* in descending order: [weight, rank] = descend(*R*);(14) Update *F**: *F** = [ *F**, *S* (:, rank(1))];(15) Delete the selected optimal feature in S: *S*(:,rank(1)) = [ ];(16) Update *j*: *j* = *j* + 1;(17) Repeat;(18) End;(19) Return the optimal subset *F** of *k* features. 

The FS [21], CFS [24], ReliefF [20], mRMR [27], and the proposed KPLS-MWMR methods can perform feature selection with supervised learning. However, the FS [21] method calculates the feature weighting according to their recognition ability without considering the redundancy between the features. The CFS [24] method chooses the features that are the most associated with the class label and have the least redundancy with the selected features in order to form the optimal feature subset, whereas the redundancy between the features is ignored. The ReliefF [20] method is based on the feature weighting between the features and the class label without considering the redundancy between the features. In addition, the mRMR [27] method can measure the connection between the features and the class labels and the redundancy of the feature subsets in line with the mutual information values but neglects the redundancy between the features in different datasets and the different association between the features and the class labels. Compared to the FS, CFS, ReliefF, and mRMR methods, the proposed KPLS-MWMR algorithm not only considers feature weighting and feature redundancy but also considers the correlation between feature weighting and feature redundancy for different datasets.

## 4. Experimental Results

For the purpose the evaluating the feature selection performance of the proposed KPLS-MWMR method, experiments were conducted by applying this method on specific datasets, and the results thus obtained were compared with those obtained by applying FS [21], CFS [24], ReliefF [20], and mRMR [27] methods on the same datasets. Here, the linear SVM classifier [32] was used as the base classifier that came with MATLAB 2016a. All the involved experiments were run on a PC comprising an Intel Core i3 2100 CPU with a maximum of 4 GB of memory, having a Windows 10 operating system, and using MATLAB 2016a software.

Three groups of experiments were performed. The first set of experiments was carried out using synthetic data to show the effect of the weight factor, α, and the redundant features on the classification performance of the proposed KPLS-MWMR method. The experimental results showed the effectiveness of the proposed feature selection algorithm. The second set of experiments was performed to analyze the learning performance of the proposed KPLS-MWMR and other feature selection methods by adopting several real datasets. Here, three evaluation criteria, namely, the classification accuracy, the kappa coefficient, and the F1-score, were adopted to distinguish the superiority of the different feature selection methods. In addition, a third set of experiments was performed for describing the sensitivity of α on the KPLS-MWMR method by analyzing the classification performance of the method at different values of α.

### 4.1. Experiments Were Performed Using Synthetic Data

This subsection presents the findings that confirmed the effectiveness of the proposed method, which was investigated by applying this method to a synthetic dataset to eliminate the redundant features. Initially, the classification performance of the proposed method after eliminating the redundant features under different weight factors was evaluated. Further, the impact of different noise levels on the classification performance of the proposed feature selection was analyzed. Finally, the classification performances of diverse feature selection methods in eliminating redundant features were investigated.

Based on a previous study [36], synthetic datasets involving three classes were constructed in this work which obeyed the normal distributions *N*(5, 1), *N*(10, 1), and *N*(15, 1), respectively. Each class of the dataset included 100 instances and three numerical features. A total of 100 noise features were added to each instance that was subjected to the normal distribution *N*(0, 0.01). 

To decrease the impact of the irrelevant and redundant features on the classification results, 10-fold cross-validation was adopted for computing the classification accuracy on the synthetic datasets. The above process was repeated 10 times, with the average value being taken as the final result. The classification performance of the KPLS-MWMR algorithm was compared with that of the other associated methods, namely, the FS, CFS, ReliefF, and mRMR algorithms.

Taking the synthetic dataset with 100 noise features as the example to illustrate the process of the proposed KPLS-MWMR algorithm on the top three features selected in detail, we observed the following: (1) The classification performance was described after eliminating the redundant features for different values of α, as displayed in Figure 2. Based on Figure 2, it was found that the classification accuracy is different for different values of α, which range from 0.1 to 0.9. When α is 0.3, the highest classification accuracy was obtained. (2) Table 1 lists the classification accuracy results of the first three features chosen by the FS, CFS, mRMR, ReliefF, and KPLS-MWMR feature selection algorithms. Except for the CFS algorithm, all other algorithms can accurately choose the real features, indicating that the class correlation is preserved in these methods for identifying the redundant features. To some extent, the classification accuracy of the FS and mRMR methods is lower compared to that of the ReliefF method. The KPLS-MWMR algorithm exhibits the highest classification accuracy. 

The feature selection performance of the KPLS-MWMR method was further tested by applying it to five synthetic datasets containing different amounts of noise features. The number of noise features was elevated from 100 to 500. Based on Figure 3, for all datasets, the classification accuracy is the highest when the number of features is three. The classification accuracy tends to increase initially with the elevating number of features but subsequently decreases as the noise features are added.

### 4.2. Experiments Performed Using Public Data

In order to efficiently and intuitively analyze the feature selection performance of the proposed KPLS-MWMR method, this method and four other feature selection methods, namely, the FS, CFS, mRMR, and ReliefF methods, were applied to eight datasets, which are often adopted for the evaluation of the performance of the feature selection methods. Ten datasets were employed to perform the experiments. Seven datasets, Ionosphere, Sonar, Musk, Arrhythmia, Madelon, LSVT, and DrivFace, can be downloaded from the UCI machine learning repository [37], and three datasets, namely, SRBCT, Lung, and Carcinom, are gene expression datasets from the Kent Ridge biomedical dataset repository [38]. These datasets were classified into training and testing datasets, where the training datasets were employed to choose the features, and the testing datasets were used for evaluating the classification performance of the different methods based on the linear SVM classifier. Additionally, the whole dataset was categorized into a 70% training set and a 30% testing set. Table 2 presents the details of the different datasets. A Gaussian kernel function, $K(x,y)=\exp(-{\left\|x-y\right\|}^{2}/s)$, with different scale or width parameters, s, was used for all the datasets [35]. The value of s for each dataset was tuned using a validation set before performing the calculations.

The classification performance was evaluated on the basis of three extensively applied metrics, namely, the classification accuracy, the kappa coefficient, and the F1-score. In order to obtain convincing experimental results, 10-fold random cross-validation was employed, and the average values of the three metrics were recorded.

(1)Classification accuracy

Classification accuracy refers to an index that can be employed to evaluate the feature selection models. Generally, the accuracy rate suggests the proportion of the samples correctly predicted by our model to all the samples that are involved in the prediction. Figure 4 presents the average classification accuracy of the diverse feature selection methods with the use of the selected features. 

(2)Kappa coefficient

The kappa coefficient [39] refers to a measure of classification accuracy. In addition, the calculation of the kappa coefficient is on the basis of the confusion matrix in an *N*-class problem. The kappa coefficient is calculated as kappa = (*p_0_* − *p_e_*)/(1 − *p_e_*), where *p_0_* represents the overall classification accuracy and *p_e_* denotes the overall classification accuracy expected by chance. Table 3 presents the kappa coefficients of the various feature selection methods obtained by applying them to the eight datasets.

(3)F1-score

The F1-score [40] is a measurement index of the classification problem, which considers the accuracy as well as recall of the classification model. Based on a maximum value of 1 and a minimum value of 0, the F1-score is considered to be a weighted average of the model accuracy and recall. Table 4 gives the F1-score of the different feature selection methods obtained by applying them to the eight datasets.

The feature selection performance of the proposed KPLS-MWMR method can be observed in Figure 4, in which the findings of this method have been compared with those of the FS [21], CFS [24], mRMR [27], and ReliefF [20] methods by applying them on the eight datasets, namely, the Musk, Arrhythmia, SRBCT, Lung, DrivFace, Carcinom, LSVT, and Madelon datasets. Here, the linear SVM classifier was employed to be the base classifier with the aim of testing the classification performance. Furthermore, the classification accuracy was obtained by 10-fold cross-validation. The *x*-axis refers to the number of selected features and the *y*-axis represents the average classification accuracy of 10 randomized experiments obtained using each feature selection method. The number of selected features was increased from 10 to 100. 

Figure 4a–h shows the performance of the various feature selection methods with the SVM classifier. In line with the results, the classification accuracy can be enhanced when the number of selected features increases. As presented in Figure 4a, the optimal weight factor, α, is 0.3 for the Musk dataset. Using the KPLS-MWMR method with 100 selected features, the highest classification accuracy can be obtained. The classification accuracy of the FS method is close to that of the mRMR and ReliefF methods with 100 selected features. The CFS method exhibits the lowest classification accuracy. In Figure 4b, the optimal value of α is 0.3 for the Arrhythmia dataset. The highest classification accuracy on the Arrhythmia dataset is obtained using the KPLS-MWMR method with 90 selected features. The classification accuracy of the FS, CFS, mRMR, and ReliefF methods with 100 selected features is close to the KPLS-MWMR method. The CFS method exhibits the lowest classification accuracy for 100 selected features. In Figure 4c, the optimal value of α is 0.9 for the SRBCT dataset. For this dataset, the highest classification accuracy is exhibited by the KPLS-MWMR method with 80 selected features. The classification accuracy of the mRMR and ReliefF methods with 100 selected features is close to the KPLS-MWMR method. The FS and CFS methods exhibit little difference in their classification accuracy in relative to the mRMR and ReliefF methods, but the number of selected features is 90. As presented in Figure 4d, the optimal value of α is 0.1 for the Lung dataset. The highest classification accuracy for this dataset is exhibited by the KPLS-MWMR method with 90 selected features. The lowest classification accuracy is exhibited by the FS method with 100 selected features. The classification accuracy of the ReliefF and mRMR methods is closer to the KPLS-MWMR method with the same number of selected features. In Figure 4e, the optimal value of α is 0.5 for the DrivFace dataset. Using the KPLS-MWMR method with 50 selected features, the highest classification accuracy is achieved for this dataset, whereas the lowest classification accuracy is exhibited by the FS method with 60 selected features. The classification accuracy of the CFS, ReliefF, and mRMR methods is similar and the number of selected features remains the same. Based on Figure 4f, the optimal value of α is 0.5 for the Carcinom dataset. Moreover, the highest classification accuracy for this dataset is obtained with the use of the KPLS-MWMR method with 100 selected features. With the increasing sample number, the rate of change of classification accuracy of the mRMR and ReliefF methods can be observed to be relatively large. With the number of features being 100, the classification accuracy of the mRMR and ReliefF methods is close to that of the KPLS-MWMR method. In Figure 4g, the optimal value of α is 0.7 for the LSVT dataset. The highest classification accuracy is obtained for this dataset using the KPLS-MWMR method with 80 selected features. Among the different feature selection methods, the classification performance of the CFS method is the lowest. In Figure 4h, the optimal value of α is 0.1 for the Madelon dataset. With the number of selected features being lower than 50, the classification performance of the proposed algorithm is poor compared to that of the ReliefF method but better than that of the CFS and mRMR methods. Nevertheless, with the number of selected features starting at 50, the KPLS-MWMR method exhibits better classification performance. With the number of selected features being 80, the highest classification accuracy can be obtained by the KPLS-MWMR method.

From the above experimental results, it is found that the classification performance of the proposed algorithm exceeds that of the FS, CFS, mRMR, and ReliefF methods. Since the datasets used in the evaluation include low- as well as high-dimensional datasets, the classification performance of the KPLS-MWMR feature selection algorithm proves its efficiency for both types of datasets. This also indicates that the performance of classification learning can be efficiently enhanced by choosing the features that are most associated with the classification by feature weight, as well as the features with the least redundancy among the selected features.

The kappa coefficients and the F1-score obtained by applying the various feature selection methods on the eight datasets are given in Table 3 and Table 4. Here, in the experiments for calculating these two metrics, the number of selected features reaches 50. According to the findings of all the experiments, it is observed that the KPLS-MWMR method exhibits comparatively good performance on most datasets. The performance of the proposed method is better relative to that of the mRMR and ReliefF methods, which indicates that the proposed algorithm exhibits advantages in preserving the features required for accurate classification. Among all the methods considered in this study, the performance of the FS method is poor, suggesting that it is insufficient for classifying samples by considering only feature weighting.

In conclusion, the KPLS-MWMR algorithm exhibits superior learning performance compared to the other feature selection algorithms considered in this study. This illustrates that the KPLS-MWMR algorithm is feasible for classification learning by introducing a weight factor,α, to adjust the ratio between the maximum weight and minimum redundancy. Accordingly, the optimal feature subset can be found by keeping the ratio of the maximum weight to the minimum redundancy in order to accurately learn its classification performance.

### 4.3. Weight Factor Sensitivity Analysis 

For different datasets, the feature evaluation criteria of the algorithm assign different weight factors to the maximum weight and the minimum redundancy, thus highlighting the characteristics of different datasets. In the KPLS-MWMR method, the weight factor, α, determines the ratio between the maximum weight and minimum redundancy in the calculation criterion, which needs to be adjusted continuously to obtain an optimal feature subset based on feature selection. 

For simplicity, we have classified the datasets into two types (low-dimensional and high-dimensional datasets) and have presented the results, according to the classification accuracy, in Figure 5. For the low-dimensional datasets, namely, the Ionosphere, Sonar, Musk, and Madelon datasets, the optimal feature number, *k*, of the datasets was set to 10. For the high-dimensional datasets, the SRBCT, Lung, DrivFace, and Carcinom datasets, only 1% of the total number of features in the datasets were selected for the feature selection experiment. In Figure 5a, the highest classification accuracy on the Ionosphere, Sonar, Musk, and Madelon datasets is obtained using the KPLS-MWMR method, and the weight factor values are 0.9, 0.7, 0.5, and 0.6, respectively. In Figure 5b, the highest classification accuracy on the SRBCT, Lung, DrivFace, and Carcinom datasets is obtained using the KPLS-MWMR method, and the weight factor values are 0.6, 0.1, 0.3, and 0.4, respectively. When the same weight factor is selected, the classification accuracy of low-dimensional and high-dimensional datasets may be different. It has been observed that: (1) For the same dataset, the classification accuracy might vary depending on the value of the weight factor α. (2) For different datasets, the classification accuracy might be different even when the same value of the weight factor α is used. (3) The best results are obtained for the value of the weight factor α that is less than 0.5 for most of the datasets used in this study.

## 5. Conclusions

To conclude, a KPLS feature selection method based on the maximum weight minimum redundancy technique was put forward to modify its classification performance. First, using the proposed KPLS-MWMR, the influence of the weight factor and the redundant features on the classification performance was investigated. According to experimental results, the effectiveness of the proposed feature selection algorithm was confirmed. Subsequently, the classification performance of the proposed KPLS-MWMR and four other feature selection methods on several real datasets was analyzed concerning three evaluation criteria, namely, the classification accuracy, the kappa coefficient, and the F1-score, to distinguish the superiority of the feature selection methods. From the comparison of the results, it was observed that the performance of the proposed KPLS-MWMR method was notably improved relative to the FS, CFS, mRMR, and ReliefF methods. Finally, the sensitivity of the weight factor was investigated using the KPLS-MWMR method by analyzing its classification performance at different weight factors. In future work, the feature selection model of high-dimensional small sample datasets and the feature selection model of classified unbalanced datasets will be proposed to solve the feature selection problem of high-dimensional small sample data and classified unbalanced data.

## Figures and Tables

**Figure 1 entropy-25-00325-f001:**
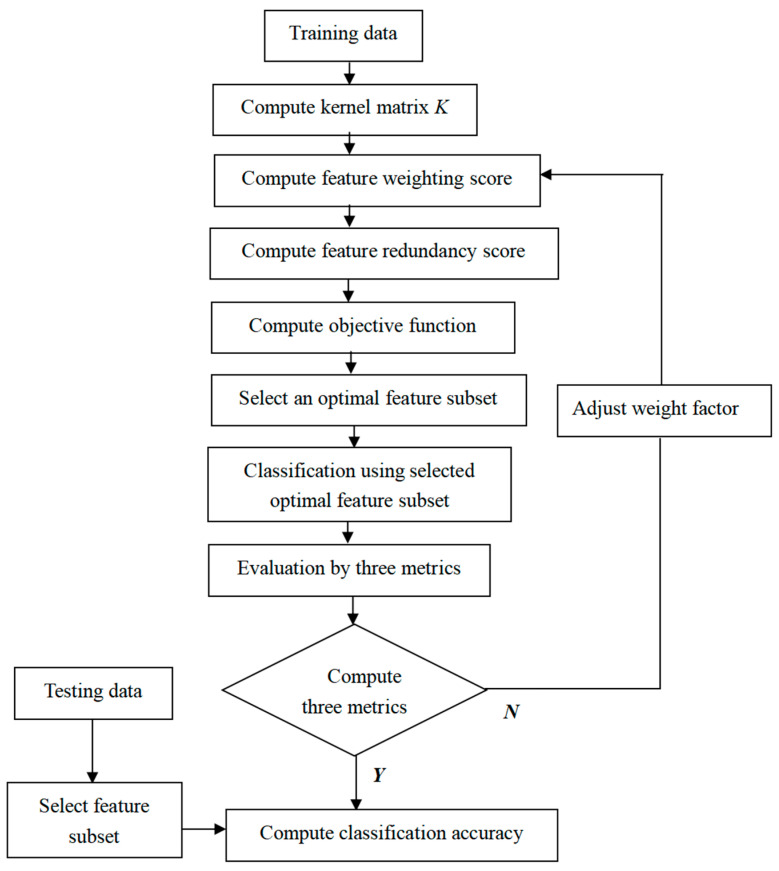
Flow chart of the KPLS-MWMR model.

**Figure 2 entropy-25-00325-f002:**
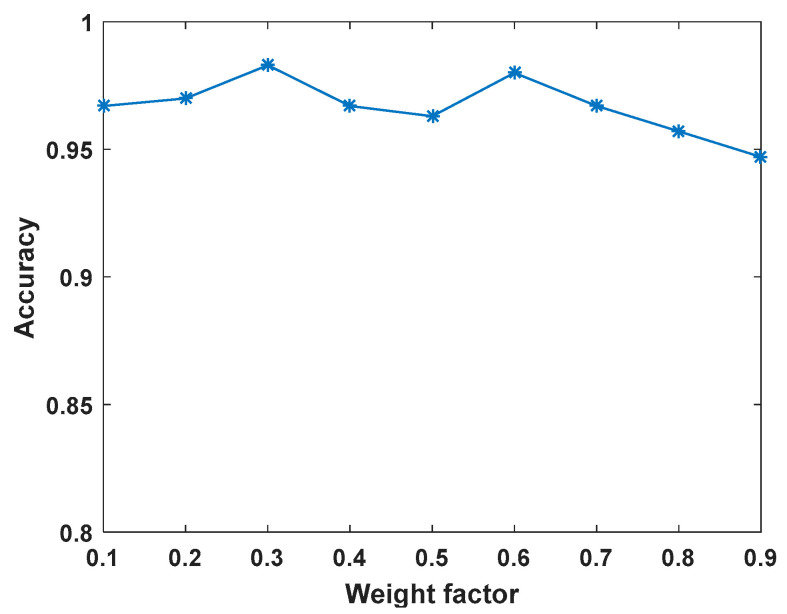
Classification accuracy using synthetic data at nine different weight factors.

**Figure 3 entropy-25-00325-f003:**
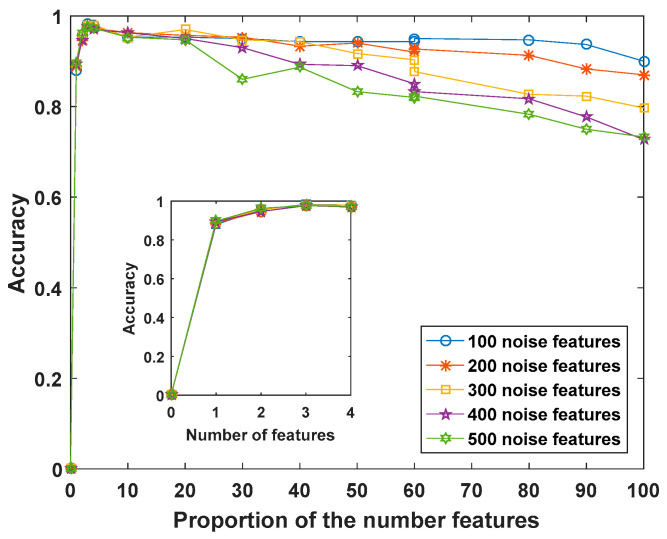
Classification accuracy of the KPLS-MWMR method obtained by applying it to five synthetic datasets.

**Figure 4 entropy-25-00325-f004:**
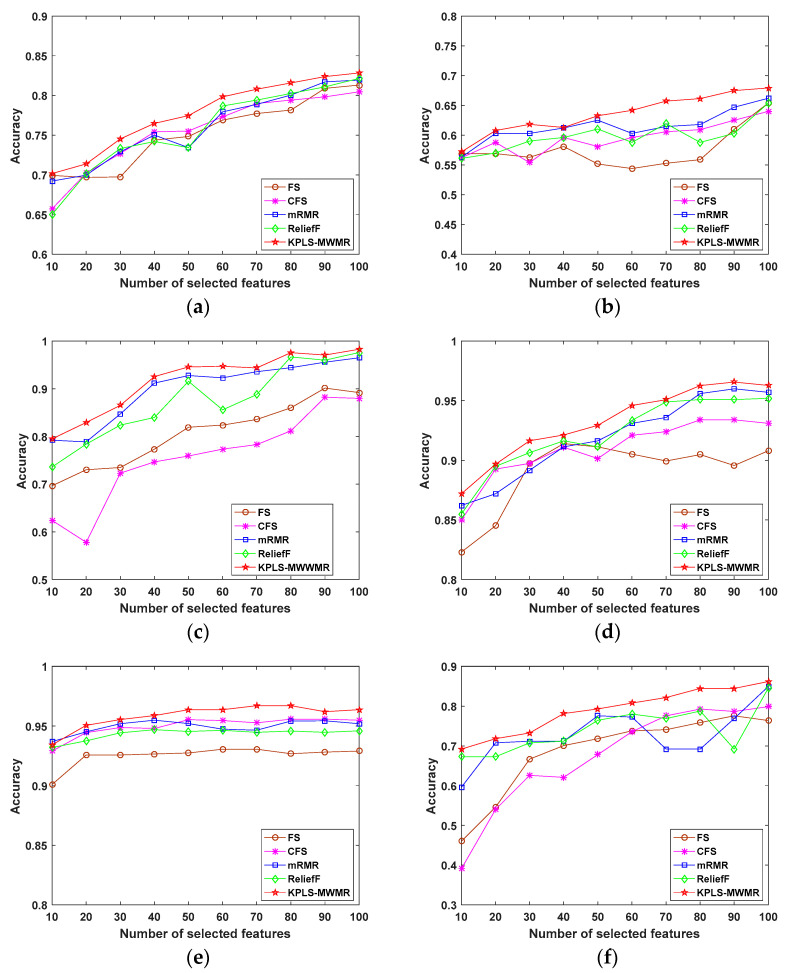
Classification accuracy of the different feature selection methods with selected features from the (**a**) Musk; (**b**) Arrhythmia; (**c**) SRBCT; (**d**) Lung; (**e**) DrivFace; (**f**) Carcinom; (**g**) LSVT; and (**h**) Madelon datasets.

**Figure 5 entropy-25-00325-f005:**
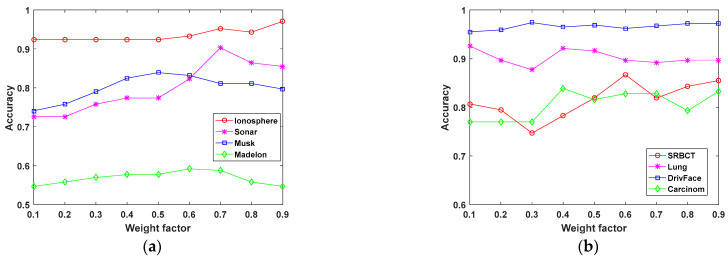
Classification accuracy of the KPLS-MWMR method with different weight factors for low-dimensional and high-dimensional datasets. (**a**) Low-dimensional datasets and (**b**) high-dimensional datasets.

**Table 1 entropy-25-00325-t001:** Top three features chosen by the different feature selection algorithms.

Algorithms	Top Three Features	Accuracy
FS	f_1_, f_2_, f_3_	0.973
CFS	f_52_, f_58_, f_3_	0.907
mRMR	f_3_, f_2_, f_1_	0.973
ReliefF	f_3_, f_2_, f_1_	0.977
KPLS-MWMR	f_1_, f_3_, f_2_	0.983

**Table 2 entropy-25-00325-t002:** Description of the ten different datasets used for evaluating the performance of the proposed KPLS-MWMR method and four other feature selection methods.

Datasets	Instances	Features	Class	Training
Ionosphere	351	34	2	246
Sonar	208	60	2	146
Musk	4776	166	2	3343
Arrhythmia	452	274	13	316
SRBCT	83	2308	4	58
Lung	203	3312	5	142
DrivFace	606	6400	3	424
Carcinom	174	9182	11	122
LSVT	126	310	2	88
Madelon	2000	500	2	1400

**Table 3 entropy-25-00325-t003:** Kappa coefficients obtained by applying the various feature selection methods to the eight datasets.

Datasets	Feature Selection Method				
	FS	CFS	mRMR	ReliefF	KPLS-MWMR
Musk	0.4807	0.5019	0.4519	0.4397	0.5143
Arrhythmia	0.1726	0.2277	0.3244	0.2772	0.3410
SRBCT	0.7510	0.6698	0.8994	0.8829	0.9383
Lung	0.8087	0.7864	0.8271	0.8876	0.8367
DrivFace	0.4295	0.7050	0.6832	0.6460	0.7062
Carcinom	0.6770	0.6295	0.7438	0.7303	0.7649
LSVT	0.5935	0.5490	0.6369	0.5714	0.6871
Madelon	0.1980	0.0450	0.0780	0.2240	0.2277

**Table 4 entropy-25-00325-t004:** F1-scores obtained by applying various feature selection methods to the eight datasets.

Datasets	Feature Selection Method				
	FS	CFS	mRMR	ReliefF	KPLS-MWMR
Musk	0.7403	0.7504	0.7259	0.7193	0.7564
Arrhythmia	0.4576	0.4851	0.4134	0.4951	0.4954
SRBCT	0.8027	0.7828	0.9300	0.9191	0.9642
Lung	0.7654	0.7947	0.8186	0.8930	0.8230
DrivFace	0.5864	0.8057	0.7891	0.7689	0.8094
Carcinom	0.6519	0.6034	0.7206	0.7099	0.7315
LSVT	0.7941	0.7705	0.8168	0.7824	0.8433
Madelon	0.5990	0.5230	0.5390	0.6120	0.6220

## Data Availability

Not applicable.

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
