# Peer review of "Kernel Partial Least Squares Feature Selection Based on Maximum Weight Minimum Redundancy"

_entropy, 2023, doi:10.3390/e25020325_

Round 1

Reviewer 1 Report

A KPLS algorithm for feature selection method on the basis of the maximum weight minimum redundancy technique was presented to improve the classification performance. The simulation results show the usefulness of the proposed scheme. I recommend the acceptance of this paper after revision.

1. There are many Kernel based methods for classification, the authors should introduce more so as to broaden the scope of this paper.

     (1)Jiaqing Qiao, and Hongtao Yin, “Optimizing kernel function with applications to kernel principal analysis and locality preserving projection for feature extraction”, Journal of Information Hiding and Multimedia Signal Processing, Vol. 4, No. 4, pp. 280-290, October 2013

      (2) De-Long Zhang, Jun-Bao Li, Li-Yan Qiao, Shu-Chuan Chu and John F. Roddick, “Optimizing Matrix Mapping with Data Dependent Kernel for Image Classification”, Journal of Information Hiding and Multimedia Signal Processing, Vol. 5, No. 1, pp. 72-79, January 2014

2. More descriptions about the Weight factor sensitivity analysis would be better.

3. Normally different amount of test data and training data will cause different performance, please describe how do you separate the data sets to what amount of train data and test data or you use some useful veridation scheme. 

Author Response

We greatly appreciate your useful comments and suggestions. Your useful comments and suggestions are very important to the perfection of the paper. We have modified the manuscript accordingly. Detailed corrections and modifications are highlighted with yellow color in the manuscript. Detailed corrections and replies are listed below point by point:

Point 1: There are many Kernel based methods for classification, the authors should introduce more so as to broaden the scope of this paper. 

Response 1: We introduced more related work about Kernel based methods for classification in section 1.

Point 2:  More descriptions about the Weight factor sensitivity analysis would be better.   

Response 2: We added more descriptions about the weight factor sensitivity analysis in section 4.3.

Point 3: Normally different amount of test data and training data will cause different performance, please describe how do you separate the data sets to what amount of train data and test data or you use some useful verification scheme.

Response 3: We adopted different verification schemes due to different experimental purposes. Experiments using synthetic data, 10-fold cross-validation was used for computing the classification accuracy on the synthetic datasets. The above process was repeated 10 times, with the average value being taken as the final result. Experiments using public data, the dataset was categorized into 70% training set and 30% testing set.The classification accuracy was obtained by 10-fold cross-validation, and the average value of 10 experimental results was recorded. The number of selected features was increased from 10 to 100. In the weight factors sensitivity analysis, for the low-dimensional datasets, the optimal feature number of the datasets was set to 10. For the high-dimensional datasets, only 1% of the total number of features in the datasets were selected for the feature selection experiment. The classification accuracy was obtained by 10-fold cross-validation, and the average value of 10 experimental results was recorded.

Please see the attachment for specific reply.

Author Response

We greatly appreciate your useful comments and suggestions. Your useful comments and suggestions are very important to the perfection of the paper. We have modified the manuscript accordingly. Detailed corrections and modifications are highlighted with yellow color in the manuscript. Detailed corrections and replies are listed below point by point:

Point 1: Please add more explicit details in the abstract (please see line 22-24) in which a summary of the most important numerical results or statistics is presented. That phrase should be something like:

“The proposed method returned the best results in 3 out of 5 cases” - or something similar, please replace the numbers 3 and 5 with actual results obtained by you. Of course, you should rephrase it in order to reflect the results obtained in the manuscript.   

Response 1: We added more explicit details in which a summary of the most important numerical results is presented in the abstract .

Point 2: Please review the minor editing errors, such as: the lines 30-31 from page 1 should be on one line, the word “repeated10 times” (line 318) should be splitted into “repeated 10 times”, and others. 

Response 2: We are very grateful to you for pointing out the editing error. We corrected these two editing errors and reviewed the full text for other editing errors.

Point 3: Please double-check if the approach in which the Algorithm 1 is written should also use numbers for the instructions of the for-loop (between lines 8 and 9), or not. If the answer is yes, then please continue the numbering after the number 8), with 9), 10), and so on.  

Response 3: We modified the writing method of Algorithm 1 in section 3.

Point 4:  Please check if it is possible to describe the characteristics of the synthetic data(s) used in the experiments using a table similar to the Table 2.

Response 4: The synthetic data is a data set with 3 classes comprising 100 examples, and each example has 103 features. In the experiment, all the synthetic data are used for 10-fold cross-validation to verify the classification accuracy of the proposed method. Therefore, we can describe the synthetic data in the experiments without using a table similar to the Table 2.

Point 5: Please extend the conclusions section with two additional future research directions.

Response 5: We added two future research directions in the conclusions section.

Point 6: Please add a high-level view chart with the main steps of the methodology proposed by you in which the main phases are presented. Please consider at least the following three phases:

  • the KPLS feature selection phase,
  • the classification phase,
  • the evaluation phase.

Response 6: We added a high-level view chart of the methodology proposed in section 3, including the use of KPLS algorithm to calculate the latent matrix, feature weighting score, feature redundancy score, objective function, selection of an optimal feature subset, classification, evaluation with three metrics, and calculation of classification accuracy of the testing datasets.

Please see the attachment for specific reply.

Round 2

Reviewer 1 Report

The authors have improved the quality of this paper, I recommend the acceptance of this paper.

Reviewer 2 Report

My comments were addressed, I recommend the acceptance of the manuscript.